# Systems with Size and Energy Polydispersity: From Glasses to Mosaic Crystals

**DOI:** 10.3390/e22050570

**Published:** 2020-05-19

**Authors:** Itay Azizi, Yitzhak Rabin

**Affiliations:** Department of Physics, and Institute of Nanotechnology and Advanced Materials, Bar-Ilan University, Ramat Gan 52900, Israel; yitzhak.rabin@biu.ac.il

**Keywords:** size and energy polydispersity, mosaic crystals, partial freezing

## Abstract

We use Langevin dynamics simulations to study dense 2d systems of particles with both size and energy polydispersity. We compare two types of bidisperse systems which differ in the correlation between particle size and interaction parameters: in one system big particles have high interaction parameters and small particles have low interaction parameters, while in the other system the situation is reversed. We study the different phases of the two systems and compare them to those of a system with size but not energy bidispersity. We show that, depending on the strength of interaction between big and small particles, cooling to low temperatures yields either homogeneous glasses or mosaic crystals. We find that systems with low mixing interaction, undergo partial freezing of one of the components at intermediate temperatures, and that while this phenomenon is energy-driven in both size and energy bidisperse systems, it is controlled by entropic effects in systems with size bidispersity only.

## 1. Introduction

Multicomponent systems of particles in which at least one of the parameters (e.g., size, interaction, etc.) varies from particle to particle, exhibit rich phenomenology compared to systems in which all particles are identical. In particular, their thermodynamic phases are quite different from one-component systems [1,2,3] and exhibit unique phenomena such as fractionation [4], i.e., phase separation into phases whose compositions are different from that of the parent phase.

Size polydisperse systems of particles with sizes which are randomly selected from various distributions (e.g., Schultz, Gaussian, uniform) were studied using molecular dynamics simulations [5,6,7,8,9,10,11,12,13]. For example, it was shown that in a size polydisperse system with Lennard–Jones interactions on the liquid–gas coexistence line, the average particle size in the liquid phase is greater than in the gas phase [5]. Another study of a size polydisperse system with a uniform size distribution has shown that in the liquid phase at constant density and temperature, increasing the polydispersity leads to slowing down of dynamics and diminishing of structural order [11].

Binary size mixtures with a single interaction parameter at high density were studied in Refs. [14,15] where both the size ratio and composition (fraction of big particles) were varied; as the size ratio is increased at low temperature, hexatic order decreases and the system undergoes a transition from a mosaic crystal to a glass (the details of the transition depend on the composition).

Previous studies by our group [16,17,18,19,20,21] have focused on energy polydisperse systems in which the parameters that characterize the strength of interactions between particles are randomly chosen from a geometric mean [16,17,19], uniform [16,18,21] or exponential [20] distributions. Using computer simulations we have shown that upon cooling, there is ordering not only of the centers of mass of the particles, but also of the identities of neighboring particles: as temperature is decreased, the system lowers its energy by arranging neighboring particles in a non-random fashion that depends on the distribution of interaction strengths and on the temperature (neighborhood identity ordering). We have also demonstrated the existence of fractionation in dilute energy polydisperse systems [19]: cooling from a gas phase results in liquid–gas coexistence (and at yet lower temperatures in solid–gas coexistence), where droplets of the condensed phase are enriched in highly interacting particles whereas the gas phase is enriched in weakly interacting ones.

As described above, size polydisperse and energy polydisperse systems exhibit some similar phenomena such as fractionation (in polydisperse hard sphere systems fractionation appears to depend on dimensionality: it has been demonstrated in 3D [12] but not in 2D [13]). Other properties of these types of systems are quite different; for example, while crystallization is suppressed in size polydisperse systems, energy polydisperse systems crystallize into periodic structures similarly to systems of identical particles. In addition, neighborhood identity ordering that was shown to exist in energy polydisperse systems has no counterpart in size polydisperse ones. In view of the above, it is interesting to explore the possibility of observing new effects in a system which combines both energy and size polydispersity. In the following we report the results of a study of a simple model of a system in which particles can have two sizes (big and small) and three interaction parameters (big-big, small-small and big-small). While models of such binary mixtures are commonly used to simulate low temperature glasses and amorphous solids (see e.g., the Kob–Andersen model [22,23,24]), other aspects of their behavior such as liquid-liquid phase separation and formation of mosaic crystals, have not been explored so far. Elucidating the qualitative features of this behavior is the objective of the present study.

The organization of this paper is as follows. In Section 2, we present the computational model and discuss the simulation algorithm. In Section 3, we present the results of our computer simulations and compare the behaviors of binary systems of large and small particles, with different choices of interaction parameters. In Section 4, we discuss the main results of this work and the new insights obtained about the physics of systems with size and energy polydispersity.

## 2. Methods

We performed Langevin dynamics simulations in Large-scale Atomic/Molecular Massively Parallel Simulator (LAMMPS) of two-dimensional systems of N=2422 particles in a square box of dimensions $L_{x}=L_{y}=L=55$ (this corresponds to number density $\rho =N/L^{2}$ = 0.80), with periodic boundary conditions in x and y directions (in NVT ensemble). Particles i and j interact via Lennard–Jones (LJ) potential:(1)$$V_{ij}\left(r\right)=4\epsilon_{ij}\left[{\left(\sigma_{ij}/r\right)}^{12}-{\left(\sigma_{ij}/r\right)}^{6}\right]$$
where *r* is the interparticle distance between particles *i* and *j* and $\sigma_{ij}=\left(\sigma_{i}+\sigma_{j}\right)/2$. The potential is truncated and shifted to zero at $r=2.5\sigma_{ij}$ (the discontinuity of the force at the cutoff distance does not affect our results since its magnitude is very small compared to the thermal force). The motion of the particles is described by the Langevin equation:(2)$$m\frac{d^{2}r_{i}}{dt^{2}}+\zeta \frac{dr_{i}}{dt}=-\frac{\partial V}{\partial r_{i}}+f_{i}$$
in which we accounted for non-hydrodynamic interactions between the particles, random thermal forces and friction against the solvent. Here ζ is the friction coefficient which we assumed to be the same for all particles independent of their size (strictly speaking, the Stokes friction coefficient increases linearly with particle radius, but since friction against the solvent is negligible compared to that due to interparticle interactions, this assumption does not significantly affect our results), V the sum of all the pair potentials $V_{ij}$ and $f_{i}$ a random force with zero mean and second moment proportional to Tζ (the temperature *T* is given in energy units, with Boltzmann constant $k_{B}=1$). All physical quantities are expressed in LJ reduced units and the simulation timestep is 0.005$\tau_{LJ}$ where $\tau_{LJ}={\left(m\sigma^{2}/\epsilon \right)}^{1/2}$ (in the following we take $\sigma =\epsilon =\tau_{LJ}=1$). The friction coefficient is taken to be ζ=0.02 which corresponds to viscous damping time $\tau_{d}=1/\zeta =50\tau_{LJ}$ that determines the characteristic transition time from inertial to overdamped motion (due to collisions with molecules of the implicit “solvent”).

In all the models studied in this work we consider a mixture of two particle sizes (effective diameters), $\sigma_{b}=1.3$ and $\sigma_{s}=0.87$ (*b* stands for big and *s* stands for small). We verified that taking the same mass ($m_{b}=m_{s}=1$) or the same mass density ($m_{b}=2.25$, $m_{s}=1.00$) of big and small particles does not affect the statistical properties of our results (not shown) and took $m_{b}=m_{s}=1$. The numbers of particles are chosen such that each type of particles has the same surface fraction; this corresponds to $N_{b}=745$ and $N_{s}=1677$. The total surface fraction is ϕ=0.66 and therefore the partial surface fraction of each component in the mixture is $\phi_{b}=\phi_{s}=\phi /2=0.33$.

The difference between the models is in the assignment of the interaction parameters. In our reference system, model R, particles of both sizes have the same interaction parameter independent of their size, $\epsilon_{bb}=\epsilon_{ss}=2$. In model A big particles have a higher interaction parameter than small particles, $\epsilon_{bb}=3$, $\epsilon_{ss}=1$. In model B big particles have lower interaction parameters than small ones, $\epsilon_{bb}=1$, $\epsilon_{ss}=3$. The values are chosen so that the unweighted average of bb and ss interactions is the same (2) in all the models. The mixing parameter $\epsilon_{mix}=\epsilon_{bs}$ that controls the interaction between big and small particles takes the same value (in the range $1\leq \epsilon_{mix}\leq 3$) in the three models. Note that large values of $\epsilon_{mix}$ are expected to promote mixing and conversely, small values of this parameter promote demixing.

The models were simulated as follows: first, particles were placed on a square lattice and the system was equilibrated at high temperature (T=10) compared to the largest value (3) of the interaction parameter during a sufficiently long time (2000$\tau_{LJ}$) in order to ensure that the fluid is completely disordered (particle positions are randomized). Then, the fluid was cooled in two steps: (1) at rate $10^{-4}1/\tau_{LJ}$ from T=10 to T=2, (2) at rate $10^{-6}1/\tau_{LJ}$ from T=2 to T=0, and measurements were performed at intermediate temperatures (this 2-step cooling was used in order to ensure structural relaxation of the system in the range T=2 to T=0).

## 3. Results

In order to study the behavior of the systems for different mixing parameters in the range $1\leq \epsilon_{mix}\leq 3$, we begin with the upper limit of this range, $\epsilon_{mix}=3$, for which strong mixing of big and small particles is expected.

The R, A and B systems with $\epsilon_{mix}=3$ undergo a transition from a homogeneous (on length scales larger than molecular size) fluid mixture (at T=10) to a homogenous glass (at T=0) that contains large voids (see the top panels in Figure 1 and Appendix A). In agreement with the expectation that the shape and composition of the interface between the condensed and the gas (void) phases are determined by surface tension minimization, in systems A and B the interfaces are enriched in weakly interacting components, i.e., small particles in the A system and big ones in the B system. Visual inspection of snapshots of the three systems taken during the process of cooling shows that the transition from a homogeneous fluid to a homogeneous glass can be characterized by the appearance of large voids that must accompany the increase of the density as the system solidifies at constant volume (not shown). Based on this criterion and on the observed change of slopes in the potential energy vs. temperature curves in Figure 2 (for $\epsilon_{mix}=3$), the glass transition temperature falls in the range 0.5–0.6 in the three systems.

Inspection of other T = 0 snapshots in Figure 1 shows that as $\epsilon_{mix}$ is decreased below 3, a gradual transition from a homogeneous glass to a solid phase in which big and small particles are progressively segregated, is observed in the 3 systems. Demixing via formation of nanocrystals of small particles, embedded in a percolating disordered network of big particles is observed at $\epsilon_{mix}=2.5$ in system B and at $\epsilon_{mix}=2$ in system R (see Figure 1). At yet lower values of the mixing parameter, this partially-ordered state is replaced by a completely ordered mosaic of big and small particle crystals (see snapshots corresponding to $\epsilon_{mix}=2$, system B and to $\epsilon_{mix}=1$, system R). The A system remains a homogeneous glass at $\epsilon_{mix}=2$ and forms a mosaic crystal at $\epsilon_{mix}=1$.

In order to understand the low temperature behavior of the three systems we note that since entropy plays no role at T=0, the above structures minimize the potential energy of the system. Thus, for sufficiently high values of $\epsilon_{mix}$, the system minimizes its energy by maximizing the number of contacts between big and small particles; conversely, for small values of $\epsilon_{mix}$ energy minimization favors maximizing the number of big-big and small-small particle contacts. A more quantitative demonstration of this effect is shown in Figure 3 where we plot the interfacial ($e_{bs}$) and pure system ($e_{bb}+e_{ss}$) contributions to the total potential energy ($e_{p}=e_{bs}+e_{bb}+e_{ss}$) as a function of $\epsilon_{mix}$, for each of the 3 systems (the $e_{p}$ value of each configuration is indicated on the snapshots in Figure 1). As expected, $|e_{bs}|$ decreases and $|e_{bb}+e_{ss}|$ increases with decreasing $\epsilon_{mix}$, as the system changes from homogeneous glass to mosaic crystal. The transition between the two states can be defined as the value of mixing parameter at which the two curves cross; this corresponds to $\epsilon_{mix}$ slightly higher (slightly lower) than 2 for R (A) systems and to $\epsilon_{mix}\approx 2.5$ for the B system. This explains the sequence of transitions in the different systems observed in Figure 1. Note that the higher value of $|e_{bb}+e_{ss}|$ in system B compared to system A results from our choice of equal volume fraction of big and small particles and from the definition of systems A and B (consequently, the number of particles with higher value of interaction parameter is larger in the B system than in the A system).

We proceed to examine the $\epsilon_{mix}=1$ case where in all systems there is a gradual demixing process from a homogenous liquid into liquid-like clusters of small and big particles that begins already during the first step of cooling at rate $10^{-4}$ to T=2 (not shown). As temperature is further decreased (see Appendix A), these clusters grow in size and eventually only two segregated large clusters of big and of small particles remain. At yet lower temperature, the three systems undergo partial freezing in which one of the components freezes while the other component remains in the liquid phase and freezes upon further cooling (complete freezing). As shown in Figure 4, in systems A and B partial freezing (of big and of small particles, respectively) takes place at $T^{*}=1.15$, which is the freezing temperature of a one-component system with $\epsilon_{ij}=3$. Complete freezing of both components (freezing of small and of big particles in systems A and B, respectively) occurs at $T^{*}=0.35$ which is the freezing temperature of a one-component system with $\epsilon_{ij}=1$ [20].

While the origin or partial freezing in systems A and B is clear (particles with higher value of the interaction parameter freeze at a higher temperature), the observation that big particles freeze before small ones in the R system is surprising since both types of particles have $\epsilon_{ij}=2$. A possible explanation may be related to the fact that system R contains more than 2 times small particles than big ones. Since entropy is proportional to the number of particles, we expect the entropy of smaller particles to dominate and to favor condensation of large particles since the resulting decrease of the entropy of the large particles is more than compensated by the increase of free space and, therefore, the entropy of the small ones. Similar entropic mechanisms are responsible for the appearance of depletion forces between colloids in colloid–polymer mixtures [25] and were shown to lead to phase separation in lattice models of hard-core binary mixtures of small and large particles [26,27]. Complete freezing in the R system takes place at the freezing temperature of a one-component system of small particles with $\epsilon_{ij}=2$ ($T_{b}^{*}=0.75$) [20].

In addition, we checked the dependence of the low temperature configurations of the three systems on the cooling method by comparing our two step cooling to fast single step cooling from T=10 to T=0 (at a rate $10^{-3}1/\tau_{LJ}$). As shown in Figure 5, fast cooling results in low temperature configurations with ramified interfaces between big and small particle crystals. The crystals contain defects i.e., isolated big particles or nanocrystals of big particles inside crystals of small particles, and vice versa. This concurs with the expectation that relaxation on length scales comparable to the size of the system is suppressed during fast cooling and that the systems become kinetically trapped in high energy states (compare the potential energy values in Figure 5 to those in Figure 1).

## 4. Discussion

In this work we used computer simulations to study dense 2d systems of particles with both size and energy polydispersity, using a simple model of a mixture of equal surface fractions of particles of two sizes, big and small, and three interaction parameters that characterize the strength of big-big, small-small and big-small interactions. We considered three representative cases: system A in which the interaction between big particles is stronger than between small particles, system B in which this situation is reversed and a reference (R) system in which there is only size but no energy polydispersity. Our goal was to find out (a) what types of phases and structures appear in those systems as they are cooled from a high temperature homogeneous fluid state down to low temperatures, and (b) how these results depend on the mixing parameter (strength of interaction between big and small particles).

In agreement with previous studies [22,23,24], we found that at high values of the mixing parameter, the three systems remain homogeneous at all temperatures and undergo a direct liquid to glass transition. At small values of the mixing parameter, lowering the temperature resulted in segregation between the two components (big and small particles), first into two liquid phases, then into one solid and one liquid phase (partial freezing) and eventually into a mosaic crystal (complete freezing). Surprisingly, all the above mentioned transitions, including partial freezing, take place not only in A and B systems where partial freezing is energy-driven (the component with larger interaction parameter freezes first), but also in system R where it is controlled by entropy. The transition between complete mixing and demixing behavior takes place at at $\epsilon_{mix}≃2$ in systems R and A and at $\epsilon_{mix}≃2.5$ in system B.

Finally, we would like to address some of the limitations of our work. The present study was carried out on a relatively small (periodic) system (2422 particles) whose size was chosen as a compromise between finite size and run time considerations. This choice of system size allowed us to do relatively short runs (10–15 h) and to explore the qualitative features of the low temperature configurations and of the different phases of the three systems, for different values of the mixing parameter. Another limitation of the present study is that we only considered attractive big-big and small-small interactions between the particles and did not observe mixed (b,s) crystals reported in ref. [28].

## Figures and Tables

**Figure 1 entropy-22-00570-f001:**
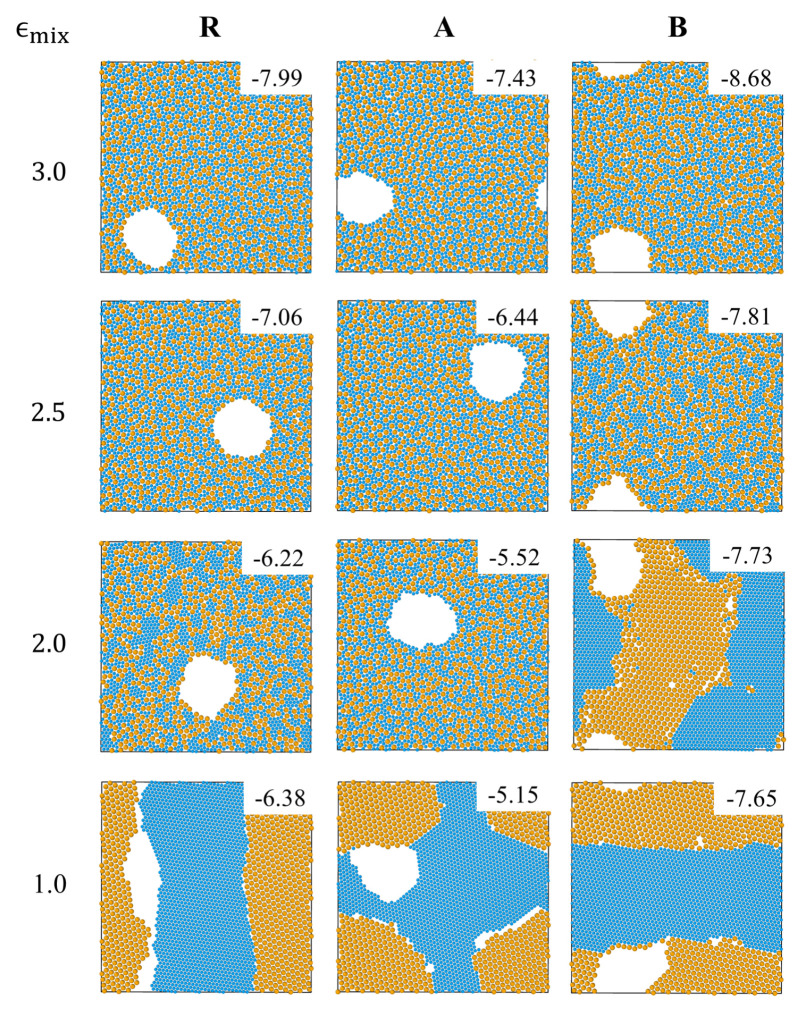
Snapshots of systems R, A and B with $\epsilon_{mix}=3.0,2.5,2.0$ and 1.0, at T=0, produced by 2-step cooling. The potential energy per particle is presented on each snapshot.

**Figure 2 entropy-22-00570-f002:**
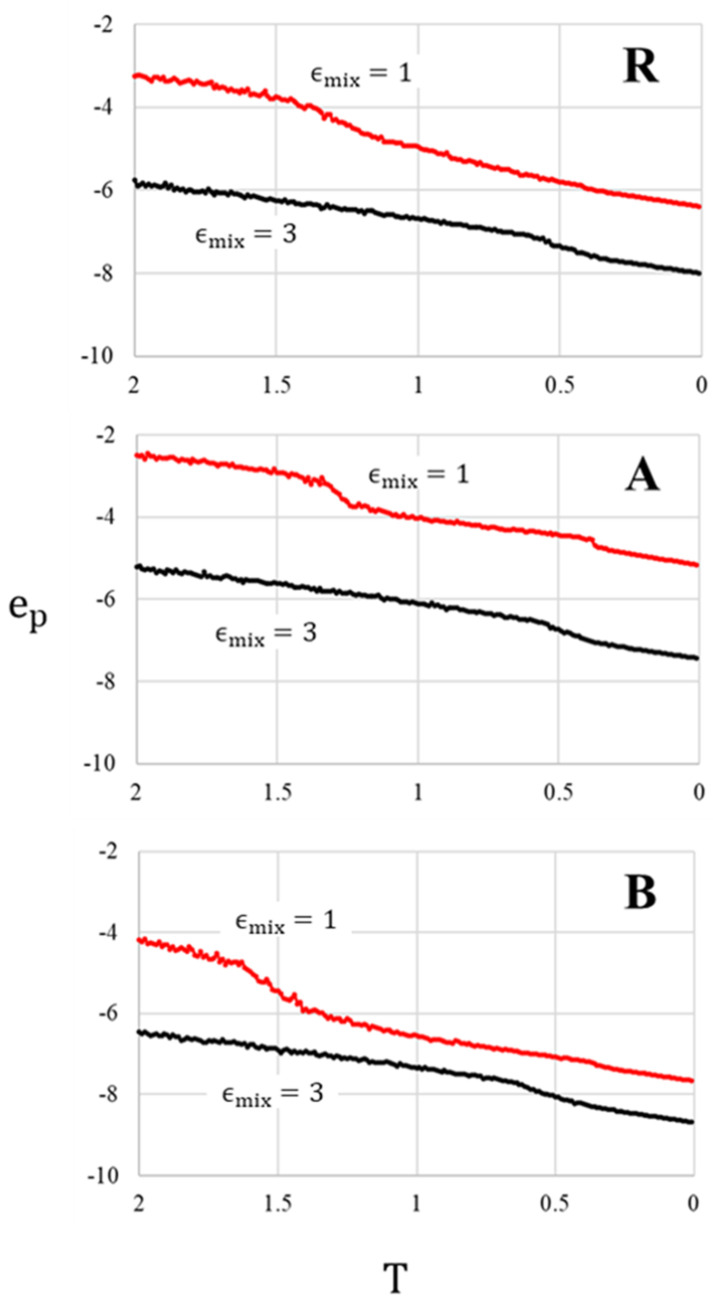
Plots of the potential energy per particle as a function of temperature, for systems R, A and B, obtained by 2-step cooling, from T = 2 to T = 0 at rate $10^{-6}1/\tau_{LJ}$.

**Figure 3 entropy-22-00570-f003:**
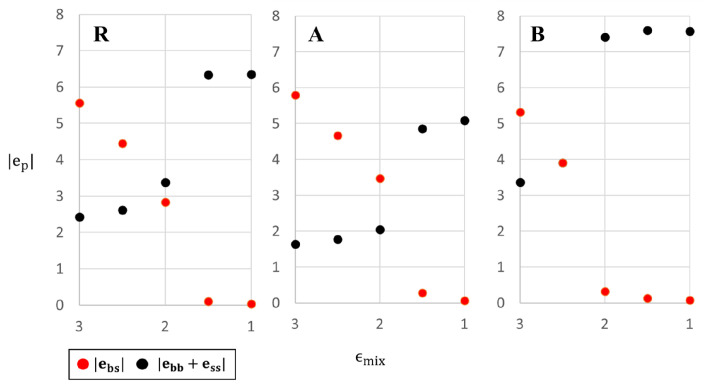
The potential energy per particle components (mixed and single-size interactions) in systems R, A and B as a function of $\epsilon_{mix}$ at T = 0.

**Figure 4 entropy-22-00570-f004:**
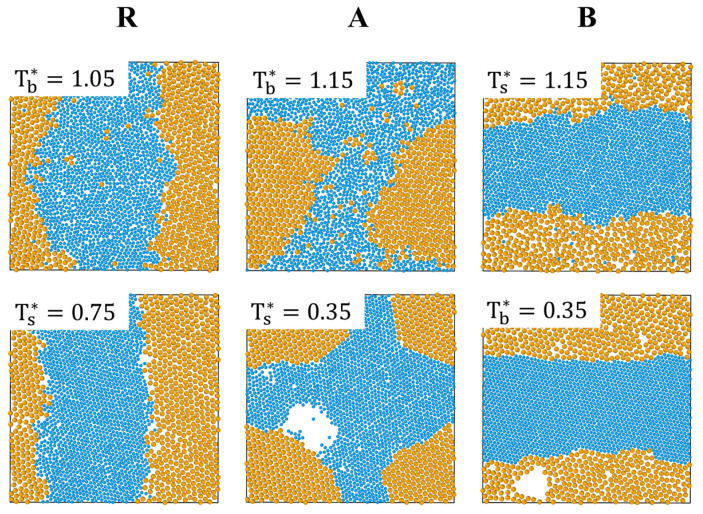
Snapshots of systems R, A and B with $\epsilon_{mix}=1$ at their partial (upper panels) and complete (lower panels) freezing temperatures. The transition temperatures are shown on the snapshots.

**Figure 5 entropy-22-00570-f005:**
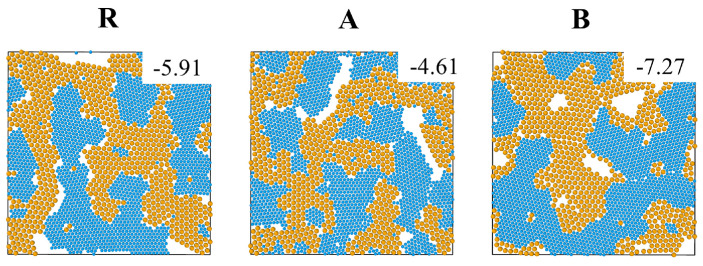
Snapshots of systems R, A and B with $\epsilon_{mix}=1$ at T = 0 produced by fast cooling ($10^{-3}1/\tau_{LJ}$). The potential energy per particle is presented on each snapshot.

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
