# Peer review of "Systems with Size and Energy Polydispersity: From Glasses to Mosaic Crystals"

_entropy, 2020, doi:10.3390/e22050570_

Round 1

Reviewer 1 Report

Review of ''Systems with size and energy polydispersity: from glasses to mosaic crystals'' by Itay Azizi  and Yitzhak Rabin.      This manuscript presents interesting observations of 2d systems of small and large particles. The authors have newly found void formation, phase separation, and glass formation simultaneously at low temperatures by changing the parameters of  simple Lennard-Jones potentials. Thus it should be published in ENTROPY. 

Author Response

We would like to thank the referee for finding our work interesting and innovative.

Reviewer 2 Report

In the manuscript by Azizi and Rabin, the authors performed Langevin dynamics simulations to study dense 2d binary mixtures of LJ particles with both size and energy polydispersity. They considered three cases: system A in which the interaction between big particles is stronger than between small particles, system B in which this situation is reversed, and a reference (R) system in which there is only size but no energy polydispersity. They found that at high values of the mixing parameter the three systems remain homogeneous at all temperatures and undergo a direct liquid to a glass. I think the results are interesting and publishable, given they can address my only concern below:

In the introduction of this work, they mentioned "if polydispersity is sufficiently high, there is phase separation into phases whose compositions are different from that of the parent phase, the phenomenon of fractionation." I agree that in 3D systems, this is true, while in 2D systems, which is the focus of this work, this may not be right. For example, for 2D polydisperse hard disks, a recent work found that there is no fractionation at even high polydispersity [Communications Physics, 2, 70 (2019)]. Therefore, I invite the authors to comment on this.

Author Response

We would like to thank the referee for drawing our attention to this work. Even though fractionation is clearly observed in our study (e.g., the observation of partial freezing which involves coexistence of small particle liquid with big particle crystal), we agree that it may not present in other polydisperse systems in 2D. We corrected this sentence accordingly and added a sentence about polydisperse hard sphere systems with appropriate references, to the introduction (in red).

Reviewer 3 Report

Although previous works on energy polydisperse systems by the current authors have been very interesting, the current paper does not present (in my view) enough new insights to warrant a stand-alone publication in Entropy. The study of binary mixtures with various parameters (size and energy) in the context of vitrification is already present in the glass literature, and more fundamental questions such as

[Other properties of these types of systems are quite different; for example, while crystallization is suppressed in size polydisperse systems, energy polydisperse systems crystallize into periodic structures similarly to systems of identical particles.]

, as worded by the authors, is not touched upon much as in other papers such as [J. Chem. Phys. 130, 224501 (2009)]. Alas, I cannot recommend publication.

Author Response

While we agree with the referee that “the study of binary mixtures with various parameters (size and energy) in the context of vitrification is already present in the glass literature”, we disagree with his conclusion that “more fundamental questions … are not touched upon as much as in other papers such as JCP 130, 22450 (2009)”. The focus of our work was to explore the consequences of correlations and anti-correlations between size and energy parameters and to compare the qualitative behaviour of systems with size and energy polydispersity to that of systems with size polydispersity only. Another goal was to explore the way in which tuning the interaction between big and small particles affects the low temperature behaviour and, in particular, the transition from a homogeneous glass to mosaic crystal. These issues are quite fundamental and are not explored in the above mentioned paper.

We agree with the referee that there are other interesting effects such the possibility of mixed (big, small) crystals that arise when only repulsive big-big and small-small particle interactions are introduced. We now mention this in the discussion and added the reference to the paper mentioned by the referee (in red).